# Patellar Tendinopathy—Does Injection Therapy Have a Role? A Systematic Review of Randomised Control Trials

**DOI:** 10.3390/jcm11072006

**Published:** 2022-04-03

**Authors:** Shibili Nuhmani, Mohammad Ahsan, Mohd Arshad Bari, Deepak Malhotra, Wafa Hashem Al Muslem, Saad Mohammed Alsaadi, Qassim Ibrahim Muaidi

**Affiliations:** 1Department of Physical Therapy, College of Applied Medical Sciences, Imam Abdulrahman Bin Faisal University, Dammam 31441, Saudi Arabia; mahsan@iau.edu.sa (M.A.); walmusallam@iau.edu.sa (W.H.A.M.); ssaadi@iau.edu.sa (S.M.A.); qmuaidi@iau.edu.sa (Q.I.M.); 2Department of Physical Education, Aligarh Muslim University, Aligarh 202002, India; arshadbari.bari@gmail.com; 3Department of Rehabilitation Sciences, School of Nursing Science and Allied Health (SNSAH), Jamia Hamdard, New Delhi 110062, India; dmalhotra@jamiahamdard.ac.in

**Keywords:** pain relief, return to sports, sports rehabilitation, conservative management

## Abstract

Injection treatment is one of the most widely used methods for the conservative management of patellar tendinopathy. The objective of this systematic review was to synthesise data from randomised control trails on the effectiveness of various injections used in the management of patellar tendinopathy. An electronic search was conducted in the Web of Science, Scopus, PubMed, and SPORTDiscus databases. To be included in the current systematic review, the study had to be an RCT conducted on human participants that investigated the effect of at least one injection treatment on the management of patellar tendinopathy. Selected studies were required to report either patient-reported outcomes or biological and clinical markers of the tendon healing. The methodological quality of the studies was appraised using the revised Cochrane risk of bias tool for RCTs (RoB 2.0). Nine RCTs on seven types of injections were included in this review, with an overall positive outcome. Pain intensity was measured in all the studies. The VISA P score was the most used outcome measure (*n* = 8). A wide variety of interventions were compared with injection therapy, including eccentric training, extracorporeal shockwave, and arthroscopy. It can be concluded that the injection treatments can produce promising results in the management of patellar tendinopathy. However, because of the limited number of studies and the disparities in the study populations and protocols, it is not possible to make a firm conclusion on the efficacy of these injection methods, and these results should be inferred with care.

## 1. Introduction

Patellar tendinopathy, which is also known as jumper’s knee, is a common musculoskeletal condition characterised by progressive activity-related anterior knee pain and tenderness at the patellar tendon [1]. This condition may lead to impaired performance in sports and activities of daily living and may negatively impact the athletic career of professional athletes. It is also seen in sedentary populations, with a prevalence of 17% among the general population [2]. Approximately 22% of elite athletes may experience symptoms of patellar tendinopathy at some point during their athletic career [3]. Up to 14% of recreational and 45% of professional jumping athletes experience symptoms of patellar tendinopathy at any given period [4]. The prevalence of patellar tendinopathy varies among sports, with a high prevalence in sports that require high-impact ballistic loading of the leg extensor muscles. The reported prevalence of patellar tendinopathy among basketball and volleyball players is 45% and 32%, respectively [3]. Cook, Khan [5] reported that more than one-third of athletes with patellar tendinopathy could not return to sports activities within six months of the injury. It is estimated that more than 50% of athletes quit active sports because of this condition [6]. Another study reported that only 46% of athletes restored their physical fitness level following patellar tendinopathy [7]. 

Despite several treatment choices, the proper management of patellar tendinopathy is still debated. A conservative approach is the first line of management. Several options are available for conservative management, such as non-steroidal anti-inflammatory drugs, eccentric training, heavy slow resistance exercises, extracorporeal shockwave therapy, and injection therapies [8].

Injection treatment is one of the most widely used approaches for the conservative management of patellar tendinopathy. A specific volume of liquid is injected in or around the affected area of the tendon to treat patellar tendinopathy. These injections are either real-time ultrasound guided or landmark guided. Various types of injections are used, including platelet-rich plasma (PRP), autologous blood, corticosteroids, and prolotherapy [9]. While a number of randomised control trials (RCTs) have been conducted on the efficacy of these injections for treating patellar tendinopathy, only one systematic review consisting mainly of case series, which was carried out in 2010, is available in the literature [10]. Thus, there is a need to conduct a systematic review of only high-quality studies to better understand the effectiveness of injection therapies for treating patellar tendinopathy. Therefore, the aim of this study is to review the effectiveness of various injections used in the management of patellar tendinopathy based on the available evidence in the literature (randomised control trails).

## 2. Methods

This systematic review was conducted as per the Preferred Reporting Items for Systematic Review and Meta-Analysis (PRISMA) guidelines. This protocol is registered in the “International Prospective Register of Systematic Reviews” (PROSPERO) under the protocol number CRD 42020199428. To define the research question, the PICOS strategy was used as shown in Appendix A.

### 2.1. Search Strategy

An electronic search was performed in the Web of Science, Scopus, PubMed, and SPORTDiscus databases with three subject headings: 1. patella (patellar, patellar tendon), 2. tendinopathy (tendinitis, tendinosis, “jumper’s knee”, “patellar tendinopathy”), and 3. injection (“platelet-rich plasma”, “platelet-rich plasma”, corticosteroid, “autologous blood”, sclerosing, “dry needling”, “hyaluronic acid”, aprotinin, “high volume injection”, prolotherapy). The Boolean operator “AND” was used between subjects, and OR was used within subject headings. The search was conducted independently by two investigators in consultation with a biomedical librarian on 18 March 2021. Details of the search are available in Appendix A. The title and abstracts were screened. Additional articles were identified from the cited references in the articles retrieved in the search. The studies retrieved from the search were reviewed by two independent reviewers, and any discrepancies were resolved by mutual understanding. To be included in the current systematic review, the study had to be an RCT conducted on human participants that investigated the effect of at least one injection treatment on the management of patellar tendinopathy. Studies were also required to report either patient-reported outcomes or biological and clinical markers of the tendon healing. Only studies with full text available in the English language and published in peer-reviewed journals were included. Reviews, editorials, letters to the editor, conference proceedings, case studies, studies comparing different types of dry needling techniques, and studies on disorders other than tendinopathy were excluded.

### 2.2. Critical Appraisal for Methodological Quality

The methodological quality of the selected studies was appraised by two independent reviewers using the revised Cochrane risk of bias tool for randomised control trials (RoB 2.0) [11]. RoB 2.0 consists of five domains for assessing the variance in randomisation, deviation from the intended intervention, missing outcome data, bias in the measurement of the outcome, and bias in the selection of the reported results [11]. The domains of these evaluations were rated as “low risk”, “high risk”, and “some concerns”.

### 2.3. Data Extraction and Synthesis

The data extraction was performed by the primary investigator, and the data were compiled into a customised data extraction form. The following data were obtained from the included studies: number and characteristics of the participants, duration of the symptoms, type of injection, intervention comparison, outcome measures, major findings, and conclusion. The following details were retrieved regarding the injections: type and characteristics of the injection, details of administration, the experience of the clinician, number of sessions, and reported complications.

## 3. Results 

### 3.1. Selection of Studies

Figure 1 presents the PRISMA flow diagram of the selection and screening process for the articles used in the study. The e-search yielded seven articles, and two additional studies were obtained from the cited references, for a total of nine studies [12,13,14,15,16,17,18,19,20].

### 3.2. Study Characteristics

A summary of the characteristics of the selected studies is available in Table 1. All the selected studies were published between 2006 and 2019. A total of 338 participants were enrolled in these nine studies. The most common type of injection studied was PRP (*n* = 4). The other types of injection used in the retrieved studies included sclerosing injection (*n* = 2), autologous blood (*n* = 2), skin-derived tenocyte-like cells (*n* = 1), dry needling (*n* = 1), corticosteroid (*n* = 1), and normal saline (*n* = 1). Pain intensity was measured in all the studies. The VISA P score was the most commonly used outcome measure (*n* = 8). The duration of the symptoms of the participants ranged from 1 to 20 months. The assessment period, time frame for pain reduction, and the number of follow-ups varied between the studies. The majority of the studies assessed either the short-term or medium-term effects of the injection techniques on outcome measures. A wide variety of interventions were compared with injection therapy, including eccentric training, extracorporeal shockwave, and arthroscopy. Five studies compared the efficacy between two injection methods [13,14,15,17,18].

### 3.3. Methodological Quality 

Table 2 presents the methodological quality of the selected studies assessed using the RoB 2.0 scale. Seven studies fell under the low-risk category, and two studies [15,16] fell under the “some concern” category. These two studies did not provide any information on concealing the allocation sequence. However, there was no apparent imbalance in the baseline.

### 3.4. Effect of Injections on Tendinopathy Symptoms

Four studies [12,13,14,15] investigated the efficacy of PRP injection on various outcome measures of patellar tendinopathy. All the selected studies reported a significant positive impact of PRP injection on patellar tendinopathy symptoms. PRP injection significantly improved the VISA P score and VAS scale at the 6- and 12-month follow-up assessments (*p* < 0.05) and in the modified Blazina scale at the 12-month follow-up assessment (*p* < 0.05) [12]. Dragoo, Wasterlain [13] also reported that PRP accelerated the recovery compared to dry needling in a short-term follow-up. However, this benefit dissipated over time. Kaux, Croisier [15] also showed a significant improvement in the VAS score, IKDC score, and VISA P score in both short-term and medium-term follow-ups irrespective of doses (*p* < 0.05 for all). Moreover, Scott, LaPrade [14] reported a similar improvement in the VISA P score, the numeric pain rating scale, and patients’ perception of changed measures on a Likert scale in short-term, medium-term, and long-term follow-ups compared to normal saline injection. 

There were two studies that investigated the effect of autologous blood on patellar tendinopathy [17,18]. Resteghini, Khanbhai [17] reported a significant improvement in the McGill pain score, VAS scale, and VISA P score in patients treated with autologous blood injection. Even though both tenocyte-like cell injection and autologous blood injection resulted in improvements in pain and knee function, patients treated with tenocyte-like cells had a rapid improvement (with an estimated difference of 8.1 VISA points) compared to patients treated with autologous blood in a six-month follow-up study [18]. Kongsgaard, Kovanen [16] reported positive short-term clinical, structural, and functional effects of corticosteroid injection, but these effects diminished in the long-term follow up.

The effects of sclerosing injection on tendinopathy were reported in two included studies [19,20]. Polidocanol was used as the sclerosing agent in both studies. The patients who received sclerosing injection reported a significant improvement in the VISA P score within the four-month follow-up period (VISA P score at baseline: 51, at 4 months: 62) [20]. Moreover, Willberg, Sunding [19] reported that, despite the beneficial effects sclerosing injection, arthroscopic shaving was a better option for the management of tendinopathy. The patients who received arthroscopic shaving had a significantly lower VAS score (at rest: 5; during activity: 12.8) than those who received a sclerosing injection (VAS score at rest: 19.2; at follow up: 41.1). The patients who received arthroscopic shaving were more satisfied (self-satisfaction score: 52.9) than those who received a sclerosing injection (self-satisfaction score: 86.8). 

### 3.5. Characteristics of Injection

The injection technique varied across the studies (Table 3). All the injections were ultrasound guided. Intratendinous injections were given in five studies [12,13,15,17,18], and peritendinous injections were given in four studies [14,16,19,20]. All the studies provided a description of how the injections were prepared. Five studies reported the experience of the clinician either in terms of the number of years of experience or by stating they were a certified or an experienced clinician [12,13,18,19,20].

## 4. Discussion

This systematic review aimed to investigate the efficacy of various injections used in the management of patellar tendinopathy. This is the first study to review the available high-quality RCTs on injection treatments for patellar tendinopathy. Nine RCTs on seven types of injections were included in this review, with an overall positive outcome. However, a well-thought-out interpretation of these results is required before reaching a conclusion.

The selected studies showed that PRP injection had encouraging results on the symptoms of patellar tendinopathy. The effects of PRP on tendinopathy are multifactorial, including platelet effects as wells as injection-related effects [21]. PRP enhances tissue repair and regeneration by delivering cytokines and other growth factors to the injured site. The injection can cause local homeostasis, which may lead to acute inflammation to enhance healing. Even though the study by Dragoo, Wasterlain [13] reported a significant beneficial effect of PRP in a shorter period, this benefit diminished over time. A possible explanation for this finding is that the participants received a single dose of PRP, and its effects may wear off over time. The researchers claimed that the PRP they used contained a high concentration of leukocytes, which is the reason why an accelerated recovery was observed in a short period in their study [13]. 

PRP is a concentrated mixture of platelets and other growth factors produced by the centrifugation of autologous blood. Several methods for preparing PRP have been reported. Each technique produces different concentrations of platelets, erythrocytes, and leukocytes. Even though PRP has been shown to produce promising results in the management of patellar tendinopathy, the inconsistency in its preparation and variations in concentrations have made this treatment controversial. Several researchers have questioned the rationale behind the use of PRP because of the lack of high-quality studies in this area. Additional high-quality RCTs are recommended to prove the superiority of PRP over other treatment methods.

The usage of corticosteroid injection to treat tendinopathy and other musculoskeletal conditions is a highly debated issue. Studies have claimed that it has short-term beneficial clinical effects, while reports of positive long-term effects are scarce. In the studies included in this review, corticosteroid injection resulted in an improvement in the short term, but its effects disappeared in the long term [16]. A relapse of the symptoms was also reported in a cohort study [22]. Steroid-induced tendon atrophy was reported in one-third of the patients who had undergone ultrasound-guided steroid injection. However, there were no degenerative changes reported in Kongsgaard’s study, as the injection was given in the para-tendon area. Corticosteroid injection is not recommended for patients with degenerative tendinopathy. It can be utilised as an adjuvant to other conservative treatments on a short-term basis in patients without tendon degeneration. Although corticosteroids provide a short positive impact, because of the relapse of the symptoms and its deteriorating effects on collagen synthesis and tendon strength, their usage for conditions such as tendinopathy should be re-examined.

Previous studies have reported neovascularisation in more than two-thirds of patients with tendinopathies [23]. These neovessels and associated nerves may be the cause of pain and other symptoms in tendinopathy. It is hypothesised that destroying these neovessels and accompanying nerves using a chemical irritant (sclerosing agents) will relieve the symptoms of tendinopathy. Polidocanol is the most commonly used sclerosing agent. It is injected into the blood vessels before they enter the tendon. Patients reported significant improvements in patellar tendinopathy symptoms after receiving a sclerosing injection in the selected studies. Sclerosing injection appears to be a promising, much-needed treatment option for patellar tendinopathy. However, further studies are needed to confirm these results.

Even though various injections showed favourable results in the studies reviewed, it is not possible to provide a firm conclusion on the efficacy of the treatments and on which method is superior to others for a number of reasons. First, there were a limited number of RCTs available on each injection method. Second, there were heterogeneities in the study populations, injection protocols, outcome measures, and concurrent treatments. One of the factors that made a comparison of the injections challenging was the heterogeneity in the outcome measures. Most of the studies used the VISA P score as the outcome measure, whereas some studies used other outcome measures, such as the VAS scale, numeric pain rating scale, and self-reported satisfaction scale. A globally accepted gold-standard outcome measure is needed to accurately compare the efficacies of various treatment methods for patellar tendinopathy. In our view, the VISA P score is the most suitable outcome measure, as it is highly reliable and specially designed for patellar tendinopathy. It has also been proven to be reliable in languages such as French, Spanish, and German. The selected studies also had differences in the population characteristics. The therapies the patients received before being recruited for the RCT varied. The mean duration of the symptoms also varied between the studies. Previous studies have shown that exercise training such as eccentric training and heavy slow resistance exercise has a major effect on the management of patellar tendinopathy. One of the selected studies in this review also reported the beneficial effects of exercise training over corticosteroid injection on pain and knee function, especially on a long-term basis [16]. Most of the studies described in this review gave exercise training and tendon loading as a concurrent treatment to patients with patellar tendinopathy. This raises the question of whether the positive effects reported in these studies should be credited to the injection or the exercise training. The exercise training provided as a concurrent treatment also varied between studies, and most of the studies did not describe it clearly. The concurrent use of exercise and tendon loading with injection treatment may be beneficial. However, it is necessary to standardise these concurrent treatments and report them.

No major adverse effects were reported in the selected studies. One patient treated with tenocyte-like cell injection had a late rupture of the tendon and progressed to surgery. One of the perceived risks of injection is that as it requires the penetration of the tendon; thus, theoretically, it may cause damage to the tendon. However, out of the nine studies, only two reported this type of complication. Most of the studies included in this review lacked a true control group in which “no intervention” or a “sham intervention” was performed. It would have been much better if the authors included a true control group, as there is a chance of self-limiting of tendinopathies. However, the authors of these studies clarified that it is unethical to create a “true control group” without any intervention.

This systematic review followed a rigorous methodology using the PRISMA statement as a guideline. Moreover, the use of the ROB 2 scale and independent reviewers helped to improve its quality. However, there are a few limitations of this study. Incomplete descriptions of the injection techniques and variation in the period of assessment, follow up, and time frame in pain reduction limited our ability to judge the impact of the effectiveness of the procedure. Moreover, it is not possible to state that all the eligible studies were included in the current review. Only articles written in the English language were considered in this review. Because of the heterogeneity of the data, a meta-analysis was not undertaken, and a descriptive narrative approach was used to analyse the data.

## 5. Conclusions

After reviewing nine RCTs, we can conclude that injection therapies can produce promising results in the treatment of patellar tendinopathy. However, because of the limited number of studies and disparities in the study populations and protocols, it is not possible to make a firm conclusion on the efficacy of these injection methods. More high-quality studies are needed to confirm the effectiveness of these injection methods and to determine which method is superior. Moreover, it will also be important to determine in which stage of tendinopathy these injections are the most effective and to monitor the side effects of these injections on a long-term basis.

## Figures and Tables

**Figure 1 jcm-11-02006-f001:**
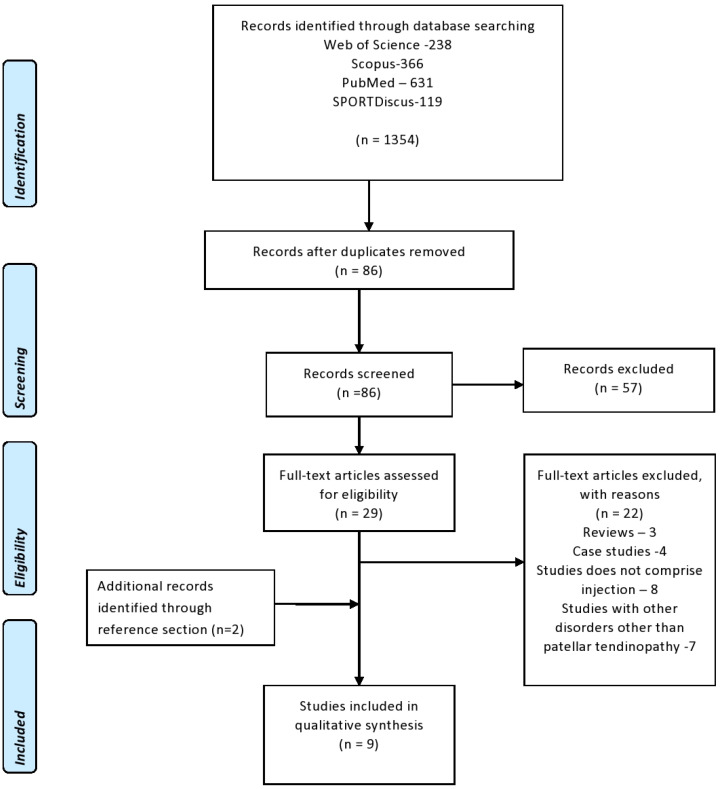
PRISMA flow diagram regarding the selection and screening of the article.

**Table 1 jcm-11-02006-t001:** Characteristics of the included studies.

Author	Sample Characteristics	Duration of the Symptoms in Months	Intervention (Injection)	Intervention Compared	Outcome Measures	Periods of Assessment	Results	Conclusion
Vetrano et al. 2013	46 athletes PRP group *n* = 23 (20 males, 3 females; age—26.9 years) ESWT group *n* = 23 (17 males, 6 females; age—26.8 years)	>6	PRP	ESWT	VISA P score VAS scale Modified Blazina scale	Baseline, 2, 6, and 12 months	**VISA P score** PRP: baseline = 55.3; 2 months = 76.2; 6 months = 86.7; 12 months = 91.3 ESWT: baseline = 56.1; 2 months = 71.3; 6 months = 73.7; 12 months = 77.6 **VAS scale** PRP: baseline = 6.6; 2 months = 3.2; 6 months = 2.4; 12 months = 1.5 ESWT: baseline = 6.3; 2 months = 3.9; 6 months = 3.9; 12 months = 3.2 **Modified Blazina scale** PRP: baseline = 6.6; 2 months = 3.2; 6 months = 2.4; 12 months = 1.5 ESWT: baseline = 6.3; 2 months = 3.9; 6 months = 3.9; 12 months = 3.2	Improvement in both groups in in short-term (2 months) and mid-term (6 and 12 months) follow up. PRP was superior to ESWT in all the clinical outcomes in mid-term follow up (6 and 12 months)
Dragoo et al. 2014	23 participants (11 men, 17 women; mean age—49.1 years) DN group *n* = 13 PRP *n* = 13	>1.5	PRP	DN	VISA P score VAS scale, Tegner activity scale, Lysholm knee scale, and Short-Form (SF-12) questionnaire	12 weeks, 26 weeks	**VISA score:** PRP- Baseline: 41, 12 weeks: 66.4, 26 weeks: 67.8 DN- Baseline: 47.4, 12 weeks: 52, 26 weeks: 83 **VAS scale** PRP- Baseline: 4.1, 12 weeks: 1.7, 26 weeks: 1.7 DN- Baseline: 3, 12 weeks: 2.3, 26 weeks:0.3	PRP accelerated recovery compared to DN but benefits dissipated over time
Scott et al. 2019	57 athletes Leukocyte-rich PRP (LR-PRP) *n* = 19 (18 males, 2 females; age—32 years) Leukocyte-poor PRP (LP-PRP) *n* = 19 (15 males, 4 females; age—33 years) Saline *n* = 19 (18 males 1 female; age—31 years)	>6	PRP	Saline	VISA P score NPRS GROC	6 weeks, 12 weeks, 24 weeks, 52 weeks	**VISA score:** LR-PRP: baseline = 49, 6 weeks = 55, 12 weeks = 63, 24 weeks = 58, 52 weeks = 58 LP-PRP: baseline = 45, 6 weeks = 57, 12 weeks = 67, 24 weeks = 71, 52 weeks = 71 Saline: baseline = 49, 6 weeks = 63, 12 weeks = 69, 24 weeks = 74, 52 weeks = 80 **NPRS** LR-PRP: baseline = 4.4, 6 weeks = 3.6, 12 weeks = 3.4, 24 weeks = 3.3, 52 weeks = 4 LP-PRP: baseline = 5.9, 6 weeks = 4, 12 weeks = 2.7, 24 weeks = 2.1, 52 weeks = 2.4 Saline: baseline = 5, 6 weeks = 3.4, 12 weeks = 2.9, 24 weeks = 3.1, 52 weeks = 42	Improvement in all the groups in patellar tendinopathy symptoms. No significant difference between the groups in all follow ups.
Kaux et al. 2016	20 patients	>3	PRP Single dose	PRP Two dose	VAS, IKDC, VISA P	6 weeks, 12 weeks	The **VAS** significantly decreased in both groups (*p* = 0.002) with no difference between the two groups. The **IKDC** score increased in both groups with values significantly higher value in single dose group (*p* = 0.0026). The **VISA-P** score increased with time in both groups (*p* = 0.0023), with no difference between the groups (*p* = 0.41).	No difference in treatment efficacy between the groups.
Kongsgaard et al. 2009	39 male athletes (age—32.4 years)	>3	Corticosteroid injections	Eccentric training Heavy slow resistance training	VAS, VISA P, tendon mechanical properties	12 weeks and half-year	**VISA-P score** Corticosteroid injection: Week 0 = 64, week 12 = 82, 6 month = 64 Eccentric training: Week 0 = 53, week 12 = 75, 6 month = 76 Heavy slow resistance training: Week 0 = 56, week 12 = 78, 6 month = 86 **VAS scale** Corticosteroid injection: Week 0 = 58, week 12 = 18, 6 month = 31 Eccentric training: Week 0 = 59, week 12 = 31, 6 month = 22 Heavy slow resistance training: Week 0 = 61, week 12 = 19, 6 month = 13	Corticostroid injection has a good short-term but poor long-term effect.
Resteghini et al. 2016	22 patients Saline group (8 males, 3 females; age—19.18 years) Autologous blood group (10 males, 1 female; age—38.91 years)	>1.5	Autologous blood	Saline	VAS VISA P SF-PMQ	12 months	**VISA P scale** Saline group: Baseline = 19.6; 1 month = 39.2; 3 month = 39.2; 1 year = 48.6 Autologous blood group: baseline = 34.1; 1 month = 50.7; 3 month = 57.7; 1 year = 62.5 **VAS scale** Saline group: Baseline = 7.9; 1 month = 4.5; 3 month = 4; 1 year = 3.3 Autologous blood group: baseline = 7.1; 1 month = 4.5; 3 month = 3.5; 1 year = 3.1 **SF-MPQ** Saline group: Baseline = 31.4; 1 month = 22.4; 3 month = 17.5; 1 year = 17.2 Autologous blood group: baseline = 22.5; 1 month = 12.6; 3 month = 10.5; 1 year = 10.7	VISA P, MPQ, and VAS scores improved significantly in both groups. There was no statistical difference between the 2 groups.
Clarke et al. 2011	46 patients (41 males, 5 females) mean age—36 years	>6	Skin-derived tenocyte-like cells	Autologous blood	VISA P	6 months	**VISA P** Tenocyte-like cell group: Baseline = 44; 6 months = 75 Autologous blood: Baseline = 50; 6 months = 70	Patients treated with tenocyte-like cells had significantly faster improvement in pain and function than those treated with autologous blood.
Willberg et al. 2011	52 athletes (49 males, 3 females) Slerosing injection group (*n* = 26; age—27.0 years) Arthroscopic shaving group (*n* = 26; age—26.6 years)	>20	Sclerosing injection	Arthroscopic shaving	VAS, Self-reported patient satisfaction	6–8 weeks, 6 months, 12 months	**VAS** Slerosing injection group: Baseline: at rest = 37.8; activity = 69.0; Follow up: at rest = 19.2; activity = 41.1 Arthroscopy group: Baseline: at rest = 44.6 activity = 76.5; Follow up: at rest = 5.0; activity = 12.8 **Self-reported patient satisfaction** Slerosing injection group: 52.9 Arthroscopy group: 86.8	Both treatments reported good clinical results. Patients treated with arthroscopic shaving showed better clinical results and patient satisfaction than those treated with sclerosing injections. Return to sports was faster in the arthroscopic shaving group.
Hoksrud et al. 2006	33 athletes (28 males, 5 females) Slerosing injection group (*n* = 17; age—25.4 years) Control group (*n* = 16; age—24.3 years)	>3	Sclerosing injection	Placebo	VISA P scale	4 months, 8 months, 12 months	**VISA score** Slerosing injection group: baseline = 51; 4 months = 62; 8 months = 70; 12 months = 72 Control group: Baseline = 53; 4 months = no change from baseline; 8 months = 79 *; 12 months = 85 *	Significant improvement in knee function and pain.

PRP—platelet-rich plasma; ESWT—extracorporeal shockwave therapy; VAS—visual analogue scale; VISA P—Victorian Institute of Sport Assessment—Patella; NPRS—numeric pain rating scale; GROC—global rating of change; IKDC—International Knee Documentation Committee scale; SF-MPQ—The Short-Form McGill Pain Questionnaire. * The control group received active treatment after four months.

**Table 2 jcm-11-02006-t002:** Risk bias assessment of the included studies.

Author	Bias Arising from Randomisation	Bias Due to Deviation from Indented Intervention	Bias Due to Missing Data	Bias in Measurement of Outcome	Bias in Selection of the Reporting Result	Overall
Vetrano, Castorina et al. 2013	Low risk	Low risk	Low risk	Low risk	Low risk	Low risk
Dragoo, Wasterlain et al. 2014	Low risk	Low risk	Low risk	Low risk	Low risk	Low risk
Scott, LaPrade et al. 2019	Low risk	Low risk	Low risk	Low risk	Low risk	Low risk
Kaux et al. 2016	Some concern	Low risk	Low risk	Low risk	Low risk	Some concern
Kongsgaard, Kovanen et al. 2009	Some concern	Low risk	Low risk	Low risk	Low risk	Some concern
Resteghini, Khanbhai et al. 2016	Low risk	Low risk	Low risk	Low risk	Low risk	Low risk
Clarke, Alyas et al. 2011	Low risk	Low risk	Low risk	Low risk	Low risk	Low risk
Willberg, Sunding et al. 2011	Low risk	Low risk	Low risk	Low risk	Low risk	Low risk
Hoksrud, Öhberg et al. 2006	Low risk	Low risk	Low risk	Low risk	Low risk	Low risk

**Table 3 jcm-11-02006-t003:** Details of the injection therapy.

Author	Type of Injection	Ultrasound Guided or Not	Detail of Administration	Details of Injection	Clinician	No. of Sessions	Complications
Vetrano, Castorina et al. 2013	PRP	Yes	Intratendinous	2 mL PRP per injection	Trained clinician	Two (one injection per week)	Local pain and discomfort in three patients on the first day of injection, which gradually subsided.
Dragoo, Wasterlain et al. 2014	PRP	Yes	Intratendinous	6 ML leukocyte-rich plasma	Board-certified radiologist	One	No complication reported.
	DN	Yes	Intratendinous	-	Board-certified radiologist	One	No complication reported.
Scott, LaPrade et al. 2019	PRP	Yes	Peritendinous	2 mL of lidocaine without epinephrine	Not reported	One	Localised patellar tendon pain for one participant.
Kaux et al. 2016	PRP	Yes	Intratendinous	6 ML PRP	Not reported	One session for one group, two sessions for the other group	No complication reported.
Kongsgaard, Kovanen et al. 2009	Corticosteroid	Yes	Peritendinous	1 mL of 40 mg/mL methylprednisolone	Physician	Two	No complication reported.
Resteghini, Khanbhai et al. 2016	Autologous blood	Yes	Intratendinous	An injection of 2 mL of 1% lidocaine, autologous blood	Two practitioners	Two	No complication reported.
Clarke, Alyas et al. 2011	Skin-derived tenocyte-like cells	Yes	Intratendinous	2 ML of tenocyte-like cells suspended in injection media (DMEM/F2)	Musculoskeletal radiologist with >12 years experience	One	One patient treated with tenocyte-like cells had a late rupture of the tendon and progressed to surgery.
	Autologous blood	Yes	Intratendinous	2 ML autologous blood plasma		One	
Willberg, Sunding et al. 2011	Sclerosing injections	Yes	Peritendinous	2 ML polidocanol (Aethoxysklerol 10 mg/mL)	Experienced ultrasonic sonographer	Three	No complication reported.
Hoksrud, Öhberg et al. 2006	Sclerosing injections	Yes	Peritendinous	2 ML polidocanol (Aethoxysklerol 10 mg/mL)	Experienced clinical assistant	Three	No complication reported.

PRP—platelet-rich plasma; DN—dry needling.

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
