# Peer review of "Patellar Tendinopathy—Does Injection Therapy Have a Role? A Systematic Review of Randomised Control Trials"

_jcm, 2022, doi:10.3390/jcm11072006_

Round 1
Reviewer 1 Report
Dear Authors,
I have read your work with great curiosity. I feel that the idea of combining a range of injection in Patellar Tendinopathy sounds interesting from a clinical point of view. However, it might pose a risk of being misinterpreted scientifically.
A precise diagnosis of patellar tendinopathy seems equally difficult. Symptoms of such a problem might originate from patella tendon, indeed. However, they could also indicate some musculo-fascial or muscular issues.
Unfortunately, the poor unclear quality of an attached fig.1 makes it unable to take the Flow Diagram into consideration.
Kind regards,
Reviewer
Author Response
Thank you for the comments
We apologize for the low quality of the figure. The original word format of the figure is attached for your kind reference.

Reviewer 2 Report
The review by Nuhmani S et al. gives a good and comprehensive overview on studies focussing on the effect of injection therapy for patellar tendinopathy.
The search of literature appears to have been thoroughly performed according to the valid PRISMA guidelines.
The outcomes of the various studies analyzed are compared in detail and limitations of these comparisons are adequately addressed in the discussion section.
Some minor remarks:
3.3. Methodological quality
Please correct the following sentence since it is not correct grammatically
"Table 2 presents the methodological quality of the selected studies was
assessed using the RoB 2.0 scale."
Table 1, p10: Hoksrud et al. 2006: the table lacks the VISA score assessment for timepoints 8M and 12M
Table 2: Please reformat the headings of the table so that they are better distinguishable
p14: 3.4. Effect of injections on tendinopathy symptoms
Please rephrase this paragraph since there is a redundancy in citation of the study of Dragoo, Wasterlain stating twice that the VISA P score was better with PRP injection than with dry needling in the short term.
"Dragoo, Wasterlain [13] also reported that PRP accelerated the recovery compared to dry needling in a short-term follow-up"
"One study compared the effects of dry needling
with PRP, and found a significant improvement in the VISA P score in both
groups[13]. In addition, the improvement in the VISA scores was significantly better with PRP injection than with dry needling in the short term."
In the same section: Here it is not clear to me which timepoints the authors compare (1M, 3M, 1year vs. 6M) and how they get to the estimated difference of 8.1 VISA points?? Please clarify
"Even though both tenocyte-like cell injection and autologous blood injection resulted in improvements in pain and knee function, patients treated with tenocyte-like cells had a rapid improvement (with an estimated difference of 8.1 VISA points) than patients treated with autologous blood in another study[18]"
Discussion, 2. paragraph:
What do the authors intend to say? Do they mean that an injection causes the disturbance of local homeostasis? Please clarify
"The injection can cause local homeostasis, which may lead to acute inflammation to enhance healing"
VISA P score: It is not clear to me why the langaguage matters in this regard. Please comment
"It has also been proven to be reliable in languages such as French, Spanish and German."
Author Response
Thank you for your comments
Please find the attached word file for the authors response

Reviewer 3 Report
Paper is well organized, tables and charts are self explanatory. References are appropriate and include previously published studies on this subject. Review is useful for practicing physicians. Therefore, please check the minor spell.
Author Response
Thank you for the comments
The manuscript has been rechecked. Grammatical and typographical errors corrected